# Robust Visual Reasoning via Language Guided Neural Module Networks

**Arjun R. Akula**[1]**, Varun Jampani**[2]**, Soravit Changpinyo**[2]**, Song-Chun Zhu**[3,4,5]
[1]UCLA Center for Vision, Cognition, Learning, and Autonomy, [2]Google Research
[3]Beijing Institute for General Artificial Intelligence (BIGAI),
[4]Tsinghua University, [5]Peking University
`aakula@ucla.edu, {varunjampani,schangpi}@google.com`
`s.c.zhu@pku.edu.cn`

## Abstract

Neural module networks (NMN) are a popular approach for solving multi-modal tasks such as visual question answering (VQA) and visual referring expression recognition (REF). A key limitation in prior implementations of NMN is that the neural modules do not effectively capture the association between the visual input and the relevant neighbourhood context of the textual input. This limits their generalizability. For instance, NMN fail to understand new concepts such as "yellow sphere to the left" even when it is a combination of known concepts from train data: "blue sphere", "yellow cube", and "metallic cube to the left". In this paper, we address this limitation by introducing a language-guided adaptive convolution layer (LG-Conv) into NMN, in which the filter weights of convolutions are explicitly multiplied with a spatially varying language-guided kernel. Our model allows the neural module to adaptively co-attend over potential objects of interest from the visual and textual inputs. Extensive experiments on VQA and REF tasks demonstrate the effectiveness of our approach. Additionally, we propose a new challenging out-of-distribution test split for REF task, which we call C3-Ref+, for explicitly evaluating the NMN's ability to generalize well to adversarial perturbations and unseen combinations of known concepts. Experiments on C3-Ref+ further demonstrate the generalization capabilities of our approach.

## 1 Introduction

Visual question answering (VQA) [11, 8, 7] and visual referring expression recognition (REF) [37, 27] are fundamental language-to-vision matching tasks that have several downstream applications such as robot navigation, image retrieval, and natural language interfaces [42, 50, 4, 3, 39]. The high-level goal of these tasks is to perform joint reasoning over visual and textual queries. In the recent years, neural module networks (NMN) [10, 20, 33] attracted increasing attention due to their superior performance on these tasks [33, 26]. Briefly, NMN models learn to parse textual queries as executable programs composed of learnable *neural modules*. Each of these modules implements a single step of reasoning (e.g. `count`, `filter`, `compare`) and are dynamically assembled to perform multi-step reasoning over text. In addition to the good performance, NMN also provide high model interpretability thanks to their transparent, hierarchical and semantically motivated architecture [45, 5, 6, 32].

Despite great success, the current NMN implementations require a large amount of training data and are less effective in generalizing to unseen but known language constructs [29, 13, 51]. For example, NMN fail to understand new concepts such as "*yellow sphere to the left*" that are constructed using a combinations of known concepts from train data such as "*blue sphere*", "*yellow cube*", and "*metallic*

35th Conference on Neural Information Processing Systems (NeurIPS 2021).

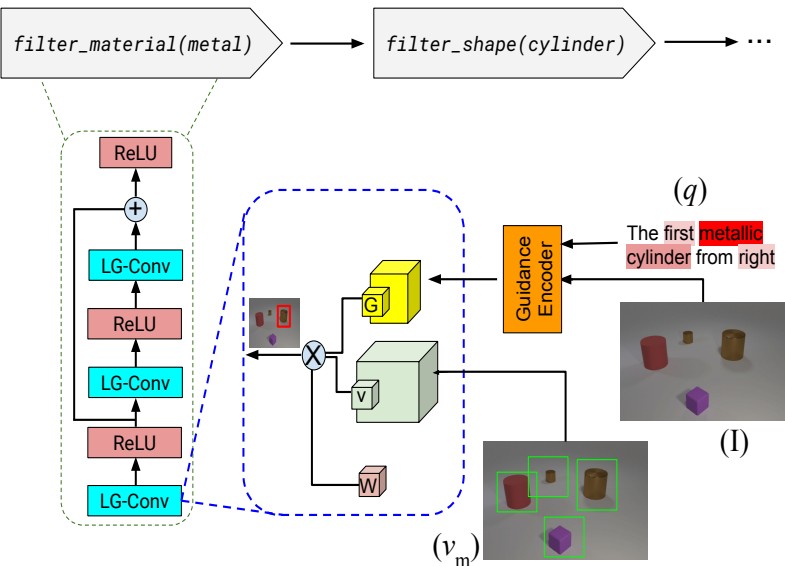

Figure 1: An example from the CLEVR-Ref+ dataset. Existing NMN implementations only provide the visual features ($v_m$) as inputs to the neural modules. In this work, we additionally condition each module on textual expression ($q$) by replacing the standard convolution layers with content adaptive convolution layers **LG-Conv** which modify the convolution by explicitly multiplying the filter weights ($W$) with a spatially varying language-guided kernel $G$. $\otimes$ denotes element-wise multiplication and $\oplus$ denotes summation.

*cube to the left*". One of the main reasons for this is that the neural modules in existing works either use a shallow, indirect language guidance [40, 19, 2] or pre-define the textual inputs in the module instantiation [26, 33], ignoring the rich correlations among the visual inputs and the relevant context from the textual inputs. For example, the neural module that filters based on the object size, "`filter_size(smallest)`", needs to localize a tiny sphere or a medium-sized sphere in the image depending on the object relationships in the expression (e.g. "*the smallest thing among the spheres*" vs. "*the metallic sphere smaller than all the large cylinders*") and the different sizes of spheres and cylinders available in its visual input. We believe that explicitly conditioning the neural modules on the joint textual and visual context helps in inferring robust visiolinguistic relationships which further enhances the compositional reasoning skills.

In this work, we address the aforementioned issues by explicitly providing the relevant objects and relationships in the textual expression to neural modules. To do this, as shown in Figure 1, we replace the standard convolution operations in the neural modules with a novel language-guided adaptive convolution operation, which we call **LG-Conv**. More specifically, the filter weights $W$ of LG-Conv are explicitly multiplied with a spatially varying language-guided kernel $G$, which allows the module to adaptively co-attend over potential objects of interest from the visual input and textual input by altering the convolution. Although content-adaptive convolutions [24, 15, 44] are used in several vision tasks, we are not aware of any prior works that does this filter adaptation using language as guidance. We propose two novel and effective methods namely, bi-salient attentional guidance (BiSAtt) network and co-salient attentional guidance guidance (CoSAtt) network to learn the guidance kernel $G$ from textual and visual inputs.

We conduct extensive experiments on VQA and REF tasks using CLEVR [25], CLOSURE [13], and CLEVR-Ref+ [33] datasets. On the recently released VQA benchmark CLOSURE [13], our approach significantly outperforms all the previous works with 11.6% improvements in accuracy. On the REF benchmark CLEVR-Ref+ [33], we outperform competing approaches by as much as +9.8% accuracy on single-referent split (S-Ref) and +4.7% on full-referent split (F-Ref), suggesting the importance of language-guidance. Most significant gains with S-Ref, which consists of only 30% of the training data in CLEVR-Ref+, demonstrate the superior generalization of our model in learning

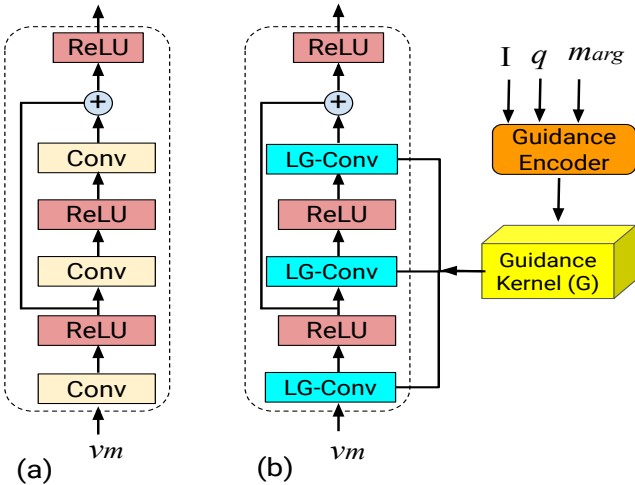

Figure 2: (a) Architecture of neural module ($m$) in existing NMN [26] consuming a visual input $v_m$; (b) Our proposed architecture replacing Conv layers with content adaptive convolution layers guided by the input image $I$, input query $q$ and parameterized textual input $m_{arg}$.

from fewer training samples[1]. We further evaluate the generalization capabilities of our approach by collecting a new dataset consisting of unseen compositions and contrasting samples for CLEVR-Ref+ benchmark [17], and call our new dataset C3-Ref+.

Our key contributions are summarized as follows:

1. We propose a novel language-guided adaptive convolution layer for NMN that guide modules in adaptively selecting informative visiolinguistic relationships and in attending to relevant objects of interest from the visual and textual inputs;
2. We demonstrate the superiority of our approach by achieving new state-of-the-art results on multiple tasks and benchmarks;
3. We introduce a new benchmark to explicitly test the model's ability to generalize to adversarial perturbations and novel compositions of concepts unseen during training. We show that our model is more robust and generalizable compared to previous approaches.

## 2   Related Work

**Multi-modal Grounding.** Early approaches for tackling grounding tasks such as VQA and REF used recurrent networks with CNNs and attention-based models to jointly understand visual and language inputs [16, 48, 34]. [8] proposed bottom-up and top-down approach to learn attention over image regions obtained from a pretrained object detector. However, these models are shown to be heavily driven by annotation artifacts in the training data [1]. Balanced and synthetic datasets such as CLEVR [25], CLOSURE [13], CLEVR-Ref+ [33] are proposed by explicitly controlling the bias and language priors. Transformers [36, 31, 47], using pretrain-then-transfer approach, have shown superior performance on these datasets. However, these models fail to learn robust visio-linguistic representations and are shown to exploit the imbalanced distribution in the train and test splits [5, 14].

**Neural Module Networks.** Neural module networks (NMN) leverage specialized modules to compute basic reasoning tasks. These modules can be assembled to perform complex and compositional reasoning [10, 25, 26, 19]. [9] proposed dynamic NMN that learns and adapts the structure of the execution layouts to the question. Recently, [26] proposed homogeneous (IEP) and generic neural modules, unlike fixed and hand-crafted neural module, in which the semantics of each neural module is learnt during training. IEP model achieves promising performance on CLEVR dataset. [33] proposed IEP-Ref by extending IEP model to CLEVR-Ref+ dataset and outperformed all the prior

---

[1]S-Ref is a subset of full CLEVR-Ref+ dataset containing expressions that refer to only a single target object in the image.

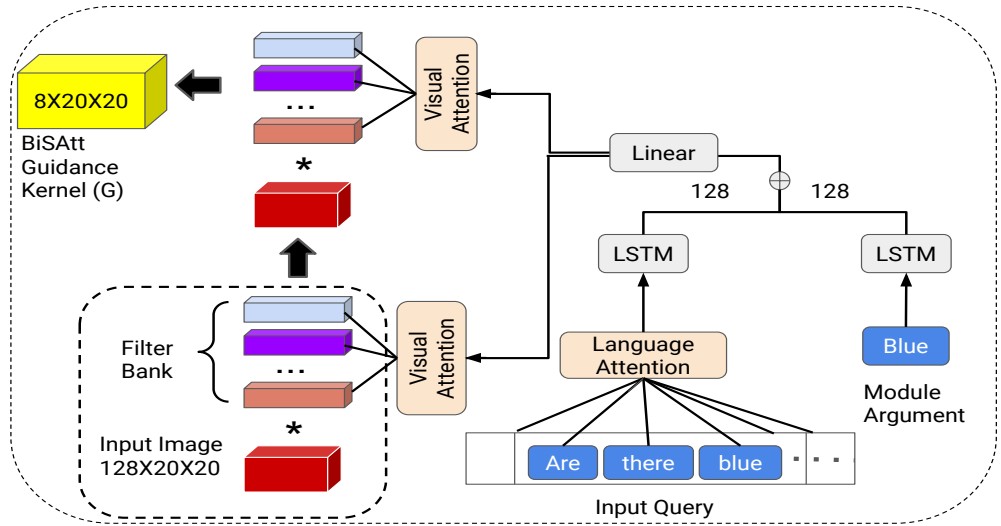

Figure 3: Bi-Salient Attentional Guidance Encoder (**BiSAtt**): In BiSAtt Encoder architecture, we first encode text inputs and then use it to learn a set of adaptive weights to linearly combine the basis filters which produces the convolution filters applied on input image.

works. Although, compositional by design, these models lack robust generalization abilities and fail to ground novel combinations of known linguistic constructs [13]. Few implementations of NMN such as FiLM [40] and N2NMN [19] condition the neural modules on textual guidance. However, the visiolinguistic context in these modules is rather shallow as they cannot jointly co-attend over potential objects of interest directly from the visual input and textual inputs. The major difference between our work and the prior works of NMN is that we explicitly condition the neural modules on the language-guidance by directly altering and adapting the convolution operation.

**Adaptive Convolutions.** There is an extensive literature on content-adaptive convolutions, in which standard 2D convolutions are generalized to high-dimensional convolutions [12, 23, 41, 18]. [23] proposes to use bilateral filtering layers inside CNN architectures. [24] proposed dynamic filter networks in which input-specific custom filter weights are predicted by a different network branch. [44] proposed pixel-adaptive convolutions which modifies the filters in a position-specific fashion. While these prior works study adaptive convolution techniques in CNN representation tasks, it has not been explored before for guiding NMN for multi-modal language-vision tasks. Closest to our work is the FiLM [40] model that learn parameters for scaling up or down the CNN activations by conditioning on the textual input. Our model differs from FiLM primarily in using language guidance that depends on learnable, local pixel and textual features.

## 3 Approach

**Problem Setup and Notation.** Given an image $I$ and a natural language query $q$ as input, our goal is to develop a NMN model that selects an answer $a \in A$ to the query from a fixed set $A$ of possible answers. We generalize this notation for both VQA and REF tasks; $q, a$ denote question and a natural language answer respectively in VQA, whereas they represent a referring expression and a bounding box of the target object respectively in REF. We represent input image $I$ as an ordered sequence of a set of image regions $R = (r_0, r_1, ..., r_N)$ and the query $q$ as the set of words $(w_1, w_2, ..., w_L)$ where $w_i$ is the $i$-th word, $N$ is the number of image regions extracted from input image $I$, and $L$ is the total number of the tokens in the input query.

Similar to [26], we use a two-stage model for generating answer: (1) Program Generation Model $p(z|q; \theta_p)$: where the query is parsed to $z$ representing the reasoning steps required to answer the query, and (2) Program Execution Model $p(a|z, I; \theta_e)$: where the predicted program $z$ is used to assemble a input-specific neural network that is composed from a set of neural modules $m$ and is executed to produce a distribution over answers.

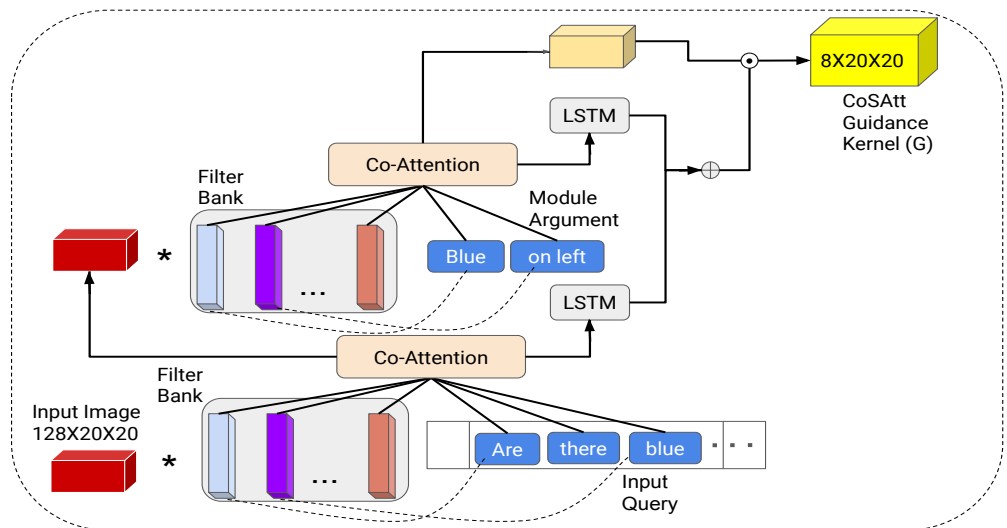

Figure 4: Co-Salient Attentional Guidance Encoder (**CoSAtt**): CoSAtt Encoder jointly attends over the input image and text inputs (early fusion) to identify co-salient regions and relationships in visual and language features that are contextually associated with each other.

As shown in Figure 2(a), the neural modules in current implementations of NMN [25, 33] typically use a standard Residual Convolution Block (RCB), consisting of convolution layers and ReLU activations. Formally, a module ($f_n$) of arity $n$ receives $n$ feature maps ($\mathbf{F_i}$) of shape $128 \times 20 \times 20$ and outputs a same-sized tensor $f_o = f_n(\mathbf{F_1}, \mathbf{F_2}, ..., \mathbf{F_n})$.

$$h_m = \text{ReLU}\left(\text{conv}_1\left(F_i\right)\right)$$
$$f_o = \text{ReLU}\left(\text{conv}_2\left(h_m\right) \oplus h_m\right) \tag{1}$$

As we can see, these modules [25, 26] are not explicitly conditioned on the input expression $q$, and therefore fails to extract robust visiolinguistic relationships. In contrast, as shown in Figure 2(b), we explicitly condition the neural modules on $q$, in addition to visual inputs, by replacing the standard convolution operations in RCB block with a novel adaptive and language-guided convolution operation, which we call LG-Conv. Also, we parameterize the module arguments, i.e., for example, we treat "`filter_material`" module as parameterized by argument ($m_{arg}$) "*rubber*" instead of as a standalone module "`filter_material[rubber]`". As a result of this parametrization, the number of a distinct set of modules used in the parameterized model reduce by 75%. We condition our LG-Conv layer on both query $q$ and the module argument $m_{arg}$ (See Figure 2). In the following, we describe the LG-Conv operation and detail its formal specification.

## 3.1 Language-Guided Convolution

Our high-level goal is to empower the neural modules to learn adapting visiolingustic features from both visual and language inputs. We achieve this by introducing novel LG-Conv layers which allows the module to adaptively co-attend over potential objects of interest from the visual and textual inputs by altering the convolution [24, 15, 44]. Formally, a standard *conv* layer in the RCB block performing a spatial convolution operation over the $n$ image pixels $P = (p_1, p_2, ...p_n)$ is given as:

$$\mathbf{p_i'} = \sum_{\mathbf{j} \in \mathbf{\Omega(i)}} \mathbf{W[c_i - c_j]p_j} + \mathbf{b} \tag{2}$$

where $\mathbf{W}$ denotes the filter weights, $\mathbf{c_i}$ denote the coordinates of the pixels in the image, $\mathbf{b}$ denotes biases, and $\mathbf{\Omega(i)}$ defines a convolution window. This convolution operation, with spatially shared weights, is agnostic to pixel features and independent of language features. As shown in Figure 1, we

modify this to depend on both pixel features and language features using a spatially varying guidance kernel $G$ as follows:

$$\mathbf{p}'_\mathbf{i} = \sum_{\mathbf{j} \in \mathbf{\Omega(i)}} \mathbf{G}(\mathbf{g_i}, \mathbf{g_j}) \mathbf{W}[\mathbf{c_i} - \mathbf{c_j}] \mathbf{p_j} + \mathbf{b} \qquad (3)$$

The spatial convolution $W$ is adapted at each pixel in the visual input using the guidance kernel $G$. Similar to [44], we represent $G$ using a fixed parametric Gaussian: $G(g_i, g_j) = \exp(-\frac{1}{2}(g_i - g_j)^T(g_i - g_j))$, where $g$ represents guidance features that we learn using the following two methods [2]: (a) **Bi-Salient Attentional Guidance (BiSAtt Encoder)**: We generate spatial guidance features using the architecture shown in Figure 3. The input image of dimensions $128 \times 20 \times 20$, the input query $q$, and the module's parametrized text argument are used in producing the guidance features. Specifically, in BiSAtt architecture, we add visual attention layers over $I$ to generate spatial guidance from non-spatial $q$. (b) **Co-Salient Attentional Guidance (CoSAtt Encoder)**: Here, we apply a joint attention over $I$, $q$, and the module argument to identify co-salient regions and relationships in visual and language features that are contextually associated with each other. The architecture is shown in Figure 4. In comparison to BiSAtt, as we show in our experiments, CoSAtt improves the relevance and interaction between objects in the image and the query. For efficient implementation, we use the same learned guidance across all the LG-Conv layers in a RCB block.

As our parametrized model require only a few number of modules, the total number of parameters in our NMN is significantly less compared to the state-of-the-art models, even though the network parameters in our parameterized module slightly increase due to the additional conv and LSTM units in the guidance encoder.

**Program Generator.** We implement program generator using an attention-based sequence to sequence (seq2seq) model with an encoder-decoder structure [46, 25] to map the input query $q$ into an executable program $z$. Both the encoder and decoder have two hidden layers with a 256-dim hidden vector. Similar to [25], we convert the decoded sequence of program functions to syntax trees (in an in-order traversal) in which each node contains a RCB module.

**Execution Engine.** The execution engine assembles a neural network using the predicted program $z$ by mapping function $f$ at each node in syntax tree to its corresponding neural module. The parent modules in the syntax tree takes the outputs from the child modules. Since we use a homogeneous architecture for designing our modules, the output generated from all modules is of same shape $128 \times 20 \times 20$. We flatten the final feature map before passing it to a multi-layer perceptron classifier, producing a distribution over all possible answers.

**Training.** During training, we find the optimal module parameters by maximizing the likelihood of the data. We optimize $p(z|q; \theta_p)$ using a policy gradient method.

$$\nabla \mathbf{J}(\theta_\mathbf{p}) = \mathbb{E}[\nabla \log \mathbf{p}(\mathbf{z}|\mathbf{q}; \theta_\mathbf{p}) \cdot \mathbf{r}] \qquad (4)$$

where $r$ is the reward and the expectation is taken with respect to rollouts of the policy. In order to enforce the network for generating the most accurate predictions, we then train the execution engine directly by maximizing $\log p(a|z, I, q; \theta_e)$ with respect to $\theta_e$.

$$\mathbb{E}[\nabla \log \mathbf{p}(\mathbf{z}|\mathbf{q}; \theta_\mathbf{p}) \cdot \log \mathbf{p}(\mathbf{a}|\mathbf{z}, \mathbf{I}, \mathbf{q}; \theta_\mathbf{e})] \qquad (5)$$

## 4 Experiments

In this section, we start by discussing the datasets and the baselines considered in evaluating our approach. We show the superiority of our approach by performing quantitative and qualitative comparisons between our method and the baselines. We then demonstrate the importance of the proposed language guided adaptive convolution through ablation studies. Finally, we present our new benchmark, C3-Ref+, and show that our approach is robust to adversarial inputs and generalizes well to novel compositions.

---

[2]We experimented more forms of guidance kernel discussed in [44], but we did not find significant improvements in NMN performance with these other kernels.

| Model | CLEVR-Dev | CLEVR-Test | CLOSURE |
|---|---|---|---|
| IEP-Ref [33] | $98.7^{\pm 0.3}$ | $97.1^{\pm 0.2}$ | $59.8^{\pm 0.4}$ |
| FiLM [40] | 96.2 | 96.9 | 58.9 |
| MAC [22] | 99.1 | 98.2 | 71.6 |
| Vector NMN [13] | 98.8 | 97.6 | 71.0 |
| NS-VQA [49] | **99.2** | **99.4** | 76.4 |
| LCGN [21] | NA | NA | NA |
| ViLBERT [35] | 95.3 | 93.0 | 51.2 |
| Visual BERT [35] | 96.0 | 92.8 | 50.6 |
| **Ours (with BiSAtt)** | $98.9^{\pm 0.2}$ | $99.2^{\pm 0.1}$ | $86.1^{\pm 0.1}$ |
| **Ours (with CoSAtt)** | $98.9^{\pm 0.1}$ | $99.2^{\pm 0.1}$ | $\mathbf{88.0^{\pm 0.2}}$ |

Table 1: Performance of our approach and baselines on CLEVR, CLOSURE benchmarks.

## 4.1 Datasets and Baselines

We evaluate our approach on both VQA and REF benchmarks. We use CLEVR [25] as the VQA benchmark, consisting of synthetically generated image and question pairs. Specifically, it consists of 100K images and 860K questions. We train our model on CLEVR train split and evaluate the performance on its val and test splits. In addition, using the model trained on CLEVR, we evaluate the performance on CLOSURE benchmark [13], consisting of novel compositions of objects and relations not seen in CLEVR train split. We then report results on CLEVR-Ref+ [33], a synthetic benchmark for referring expressions. It contains nearly 0.8M referring expressions of which 32% of expressions refer to only a single object (*Single-referent*) and 68% refer to more than one object (*Multi-referent*). We refer to the full dataset as F-Ref and the single-referent subset as S-Ref.

We compare the performance of our approach against several NMN baselines such as FiLM (Feature-wise Linear Modulation), MAC [22], IEP-Ref [33] VectorNMN [13], and non-NMN baselines such as NS-VQA [49], LCGN [21], ViLBERT [35], and VisualBERT [30].

| Model | S-D | S-T | F-D | F-T |
|---|---|---|---|---|
| IEP-Ref | $49.8^{\pm 0.1}$ | $51.5^{\pm 0.6}$ | $80.5^{\pm 0.2}$ | $78.2^{\pm 0.3}$ |
| FiLM | 44.9 | 46.7 | 76.5 | 75.7 |
| MAC | 46.3 | 49.2 | 81.3 | 77.4 |
| Vector NMN | 48.3 | 53.5 | 83.2 | 77.1 |
| NS-VQA | 51.5 | 52.9 | 82.5 | 79.6 |
| LCGN | 46.8 | 48.0 | 77.0 | 74.8 |
| ViLBERT | 42.4 | 44.3 | 69.3 | 68.7 |
| Visual BERT | 41.7 | 43.2 | 69.8 | 63.2 |
| **Ours (with BiSAtt)** | $61.1^{\pm 0.3}$ | $59.7^{\pm 0.2}$ | $87.2^{\pm 0.3}$ | $83.5^{\pm 0.2}$ |
| **Ours (with CoSAtt)** | $\mathbf{62.3^{\pm 0.1}}$ | $\mathbf{63.3^{\pm 0.1}}$ | $\mathbf{89.1^{\pm 0.2}}$ | $\mathbf{84.3^{\pm 0.3}}$ |

Table 2: Performance of our language-guided NMN models and state-of-the-art models on S-Ref Dev (S-D), S-Ref Test (S-T), F-Ref Dev (F-D) and F-Ref Test (F-T).

| Model | CLS | S-T | F-T |
|---|---|---|---|
| **Ours** | **88.0** | **63.3** | **84.3** |
| C1 Ours-L+G | 62.1 | 51.6 | 77.8 |
| C2 Ours-L-G | 61.5 | 52.1 | 75.2 |
| C3 Ours+L+(G w/o $I$) | 80.2 | 57.7 | 79.9 |
| C4 Ours+L+(G w/o $q$) | 78.8 | 54.1 | 76.2 |
| C5 Ours+L+(G w/o $m_{arg}$) | 82.1 | 61.7 | 80.9 |

Table 3: **Ablations.** Performance of our model with and without LG-Conv layer (L) and CoSAtt encoder (G) on CLOSURE (CLS), S-Ref Test, and F-Ref Test.

## 4.2 Implementation Details

Similar to [25], we use 18K ground-truth programs to train the program generator (PG). We train PG and the execution engine using Adam [28] with learning rates 0.0005 and 0.0001, respectively. Our PG is trained for a maximum of 32K iterations, while EE is trained for a maximum of 450K iterations. We employ early stopping based on validation set accuracy. While reporting accuracies on S-Ref test split, we use the model trained on S-Ref train split. We repeat the experiment 5 times on each benchmark and report the mean/variance on each of them.

## 4.3 Evaluation

Table 1 and Table 2 show results in comparison with the baselines. We find that our model outperforms all prior work on CLOSURE, and CLEVR-Ref+ benchmarks, while showing on-par performance on CLEVR test split. This demonstrates the effectiveness of the proposed language guided convolutions in capturing visiolinguistic relations and contextual dependencies from the longer CLEVR-like expressions. In particular, we achieve +11.6% in accuracy on CLOSURE test split compared to the best prior model Vector-NMN, indicating that our model generalizes well to unseen compositions. The multi-modal transformer based approaches ViLBERT and VisualBERT performed poorly on both CLOSURE and CLEVR-Ref+, probably due to the mismatched image distribution in pre-training (with conceptual captions [43]) and fine-tuning. Our model improves the accuracy on CLEVR-Ref+ test splits by 9.8% on S-Ref and 4.7% on F-Ref, compared with the current state-of-the-art method IEP-Ref. Significant gains on S-Test also suggest the superior generalization skills of our model in learning from fewer training samples. Relatively more improvements with CoSAtt encoder compared to BiSAtt encoder shows that early fusion of image and text features facilitate in generating more robust guidance kernel.

To gain better insight into the relative contribution of the design choices we made, we perform experiment with the following five ablated models:
**C1: Conv vs. LG-Conv (L).** We investigate the contribution of the proposed content adaptive convolution layer in the RCB block by replacing LG-Conv layer with standard convolution. In this setting, we use guidance (G) from CoSAtt encoder for directly scaling up or down the CNN activations in the RCB block.
**C2: Conditioning on CoSAtt Guidance (G).** In this ablation, we use LG-Conv layers but skip the CoSAtt encoder to verify the importance of module level conditioning on the interaction between image and text features. We instead only pass the module argument ($m_{arg}$) as guidance to the LG-Conv layer.
**C3: CoSAtt w/o. Image (I)** We encode guidance using only input query $q$ and the module argument to test the importance of conditioning on image $I$ in the CoSAtt encoder.
**C4: CoSAtt w/o. Query (q)** We encode guidance using only input image to test the importance of conditioning on input query $q$ in the CoSAtt encoder.
**C5: CoSAtt w/o. Module Arg ($m_{inp}$)** In this variant, we keep $q$ and $I$, but skip $m_{arg}$ in the CoSAtt encoder.

The results are shown in Table 3. As we can see, all the above five variants underperform, confirming the importance of our proposed content-adaptive convolutions and guidance kernel. Results show that module argument in the CoSAtt guidance has less significant effect compared to other components, suggesting that our model is able to infer the semantic context of the module. Figure 5 illustrates the qualitative differences of `filter_color(red)` module trained using IEP-Ref and our approach. With IEP-Ref, the model selects all red objects from the image, ignoring the context in the expression. On the other hand, our approach correctly locates objects based on their contextual relevance.

## 4.4 The C3-Ref+ Dataset

Following [17, 38, 13], we construct a new benchmark, C3-Ref+, to critically examine the generalization capabilities of NMN in grounding out-of-domain (o.o.d) referring expressions - a fundamental expressive power of human intelligence. Specifically, C3-Ref+ consists of two kinds of samples constructed using S-Ref split of CLEVR-Ref+ dataset: (a) **Novel Compositions**, consisting of samples that evaluate the model on combinations of objects and their spatial relationships not seen in S-Ref train split; and (b) **Contrast Sets**, consisting of samples that help in exposing model brittleness by probing a model's decision boundary local to examples in the S-Ref test set. Table 4 shows few examples.

In constructing novel compositions of samples, we first extract all the combinations of simple and complex noun phrases (e.g. "*The metallic object*", "*The metallic object to the left*") from the S-Ref train split and then manually construct new expressions using the unseen pairs of these phrases. To construct a contrast sample, we manually perturb the semantics of various parts of the referring expressions in the S-Ref test split such that the ground-truth referent object changes. For example, we modify the expression *first one of the tiny rubber thing from left* to *first one of the tiny metallic thing from right*. We verify and validate the correctness of the collected samples and their ground-truth annotations using three human annotators. The annotations that are not consistent among the three

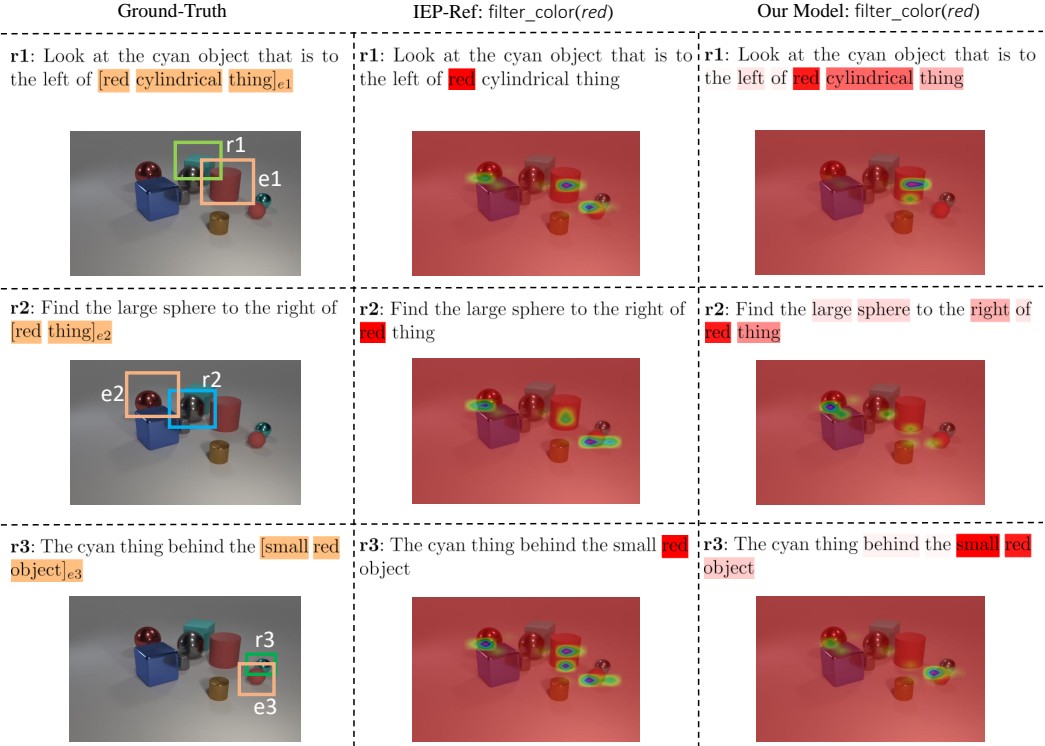

Figure 5: Qualitative examples showing attention heatmaps of `filter_color(red)` module outputs trained using IEP-Ref and our model on CLEVR-Ref+ dataset. $e1$, $e2$, $e3$ highlight the red objects that are referred in the input expressions $r1$, $r2$, and $r3$, respectively.

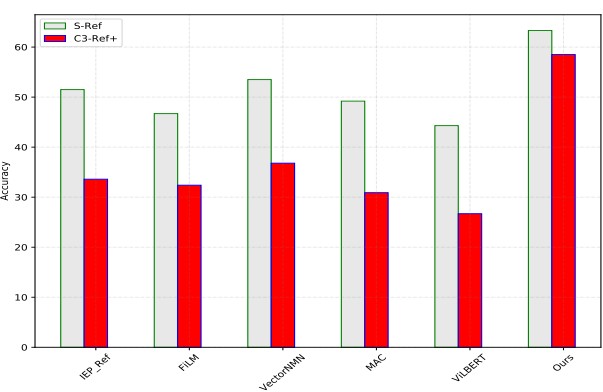

Figure 6: Accuracy of models on S-Ref and C3-Ref+

human annotators are removed from the final split [3]. We collected a total of 1412 expressions spanning 980 images with a vocabulary of 56. The average length of expressions is 18.3. More than 60% of our collected expressions have 2 or 3 relations such as "in the front" and "from the right". As shown in Figure 6, performance of state-of-the-art models drop by up to 18% on C3-Ref+. Our proposed method shows least drop ($<$5%) in performance indicating its superiority in grounding new unseen compositions and adversarial perturbations.

---

[3] We also perform an additional round of filtering to remove inconsistent samples that could point to more than one object in the image.

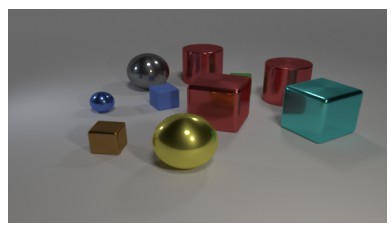 **Original:** Any other yellow metal thing(s) of the same size as the first one of the cyan metal thing(s) from right
**C3-Ref+:** Any other tiny thing(s) of the same size as the first one of the brown metal thing(s) from left

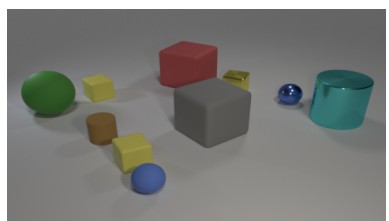 **Original:** The shiny thing(s) that are left of the first one of the small sphere object(s) from right and behind the fourth one of the rubber object(s) from front.
**C3-Ref+:** The shiny thing(s) that are right of the first one of the small cubical object(s) from right and behind the fourth one of the rubber object(s) from front.

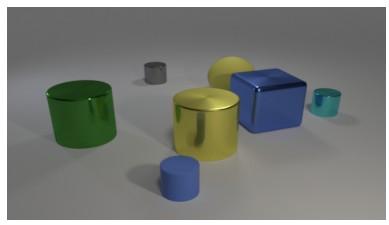 **Original:** Find matte thing that is on the left side of the cyan object that is behind the second one of the metallic object(s) from front; The last one of the object(s) from front that are behind it.
**C3-Ref+:** Find matte thing that is on the left side of the cyan object that is behind the second one of the metallic object(s) from front; The second one of the object(s) from right that are in front of it.

Table 4: Random examples of contrast sets in C3-Ref+ and their original annotations in S-Ref.

# 5   Conclusion

Neural module networks (NMN) are widely used in language and vision tasks. We show that explicitly conditioning neural modules on the language guidance through adaptive convolutions improve their grounding and generalization abilities, achieving a new state-of-the-art results on the visual question answering and visual referring expression recognition tasks. Our analysis on CLOSURE, CLEVR-Ref+ and a new compositional and contrastive split C3-Ref+ demonstrate that our proposed method enhances NMN' ability in adaptively selecting and exploiting informative visiolinguistic relationships.

# 6   Broader Impact

This work contributes to improve the joint understanding of image and textual content, which in turn is a very important component in several vision and language grounding tasks. Our research can promote the development of a multi-modal interaction system and facilitate people's daily lives. This work does not present any foreseeable societal consequence.

# 7   Funding Disclosure

*1. Funding (financial activities supporting the submitted work):*
None
*2. Competing Interests (financial activities outside the submitted work):*
None

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
