# Robust Visual Reasoning via Language Guided Neural Module Networks

In this supplementary material, we begin by providing more details on CLEVR-Ref+ full referent (F-Ref) and single referent (S-Ref) splits. We then provide a detailed comparison on the total number of parameters in our implementation and baselines. Next, we present additional details on our implementation (e.g., initialization & training, hyper-parameters) to supplement Section 4 of the main paper. We then present random examples from our proposed C3-Ref+ dataset to supplement Section 4.4. Finally we provide additional results and analysis to supplement Section 4.5 of the main paper.

## A    CLEVR-Ref+ splits

CLEVR-Ref+ [5] is a synthetic diagnostic benchmark for visual referring expression recognition task. There are nearly 800,000 expressions of which 32% of expressions refer to only a single object (*Single-referent*) and 68% refer to more than one object (*Multi-referent*). In this paper, we refer to the full dataset as F-Ref and the single-referent subset as S-Ref. Detailed statistics of the splits are presented in Table 1.

|  |  | F-Ref | S-Ref |
|---|---|---|---|
| Train Set | #Expressions | 628915 | 200313 
(32% of F-Ref) |
|  | #Images | 70000 | 62016 |
| Val Set | #Expressions | 69879 | 22256 |
|  | #Images | 6500 | 5200 |
| Test Set | #Expressions | 149741 | 47731 |
|  | #Images | 15000 | 13534 |

Table 1: F-Ref and S-Ref splits in CLEVR-Ref+ benchmark.

## B    Module Parameters in NMN

In this section, we compare the parameters of our language-guided NMN implementation with the state-of-the-art NMN models. Our NMN implementation extends **IEP-Ref** [5], the current state-of-the-art neural module network (NMN) model for the CLEVR-Ref+ dataset. Similar to IEP-Ref, we use a generic design of neural module architecture adapted from IEP [2]. For our experiments, we used the IEP-Ref implementation available at the link `https://github.com/ruotianluo/iep-ref`. The neural modules take either two visual inputs (binary modules) or one visual input (unary modules). In the original IEP-Ref implementation, there are total 60 distinct modules in IEP-Ref. As we discussed in Section 3 of the main paper, we parametrize the module arguments, i.e. for example, we treat "`filter_material`" module as parametrized by argument "`rubber`" instead of as a standalone module "`filter_material[rubber]`". As a result of this parametrization, the

| Model | #Parameters (per module) |
|---|---|
| IEP-Ref | 442,752 |
| FiLM | 590,720 |
| Vector-NMN | 443,122 |
| Language-Guided NMN | 599,341 |

Table 2: Total number of parameters per each neural module in the state-of-the-art NMN models and our proposed language-guided NMN model.

| | Modules |
|---|---|
| Unary | `Filter_Shape`, `Filter_Color`, `Filter_Material`, `Filter_Visible`, `Filter_Size`, `Filter_Ordinal`, `Unique`, `Relate`, `Same_Size`, `Same_Shape`, `Same_Color`, `Same_Material`, `Scene` |
| Binary | `Intersect`, `Union` |

Table 3: Distinct Modules in our Language-Guided NMN implementation

number of a distinct set of modules used in the parametrized model drop to 15. We compare the parameters per module of all baseline NMN models and our proposed model in Table 2.

## C    Implementation Details

We start with the baseline implementation at `https://github.com/ruotianluo/iep-ref` and modify it by incorporating language-guided neural modules. We use GloVe to obtain the word embedding (dimension = 300) of each word in the textual input. We used 18K ground-truth programs to train the program generator (PG). When training, we first train our PG and use it as a fixed module for training the execution engine (EE). We train PG and the execution engine using Adam [3] with learning rates 0.0005 and 0.0001, respectively. Our PG is trained for a maximum of 32,000 iterations, while EE is trained for a maximum of 450,000 iterations. We employ early stopping based on validation set accuracy. While reporting accuracies on S-Ref test split, we use the model trained on S-Ref train split. In reporting the model performance, we repeat an experiment 5 times on each benchmark and report the mean/variance on each of them. We train the baselines ViLBERT [6][1] and VisualBERT [4] on 8 Tesla V100 GPUs with a global batch size of 512. For all the other baselines and our model, we train on 2 RTX 2080ti GPUs with a global batch size of 16.

## D    More Examples from C3-Ref+

We construct a new benchmark, C3-Ref+, to critically examine the generalization capabilities of NMNs in grounding out-of-domain (o.o.d) referring expressions. Specifically, C3-Ref+ consists of two kinds of samples constructed using S-Ref split of CLEVR-Ref+ dataset: (a) *Novel Compositions*, consisting of samples that evaluate the model on combinations of objects and their spatial relationships not seen in S-Ref train split. Table 6 provide examples of novel compositions in C3-Ref+; and (b) *Contrast Sets*, consisting of samples that help in exposing model brittleness by probing a model's decision boundary local to examples in the S-Ref test set. Table 7 provide examples of contrast sets in C3-Ref+.

## E    Additional Results

In this section, we provide more results comparing the performance of our model with baselines. Specifically, we analyze the model's performance in terms of filtering the objects based on the attributes color, size, shape, material, ordinality, and visibility. Table 4 and Table 5 show the performance of our language-guided NMN and baseline NMN models on F-Ref and S-Ref test splits respectively. We compare the output of neural modules with ground-truth functional program annotations. Results show that our approach significantly outperforms baselines. In particular, we

---

[1]ViLBERT 8-Layer model at the link `https://github.com/jiasenlu/vilbert_beta`

| Model | filter_color | filter_size | filter_shape | filter_ordinal | filter_material | filter_visible |
|---|---|---|---|---|---|---|
| IEP-Ref [5] | **89.1** | 91.7 | 88.3 | 64.2 | 93.5 | 87.2 |
| FiLM [7] | 86.6 | 92.0 | 90.1 | 66.3 | 87.1 | 82.0 |
| Vector NMN [1] | 86.0 | 93.1 | 86.5 | 60.2 | 89.0 | 88.2 |
| NS-VQA [8] | 89.0 | 94.1 | 89.8 | 66.1 | 87.2 | 89.4 |
| **Ours (with BiSAtt)** | 88.8 | 94.2 | 88.6 | 73.1 | 95.3 | 92.5 |
| **Ours (with CoSAtt)** | 88.9 | **95.6** | **90.3** | **74.3** | **95.3** | **92.6** |

Table 4: Performance of neural modules in our language-guided NMN implementation vs. state-of-the-art NMN models (on F-Ref test split).

| Model | filter_color | filter_size | filter_shape | filter_ordinal | filter_material | filter_visible |
|---|---|---|---|---|---|---|
| IEP-Ref [5] | 63.1 | 60.0 | 55.1 | 38.8 | 53.7 | 49.1 |
| FiLM [7] | 60.9 | 58.7 | 50.8 | 32.4 | 50.1 | 44.0 |
| Vector NMN [1] | 61.4 | 59.3 | 52.5 | 33.0 | 50.1 | 44.7 |
| NS-VQA [8] | 63.8 | 61.3 | 54.2 | 38.9 | 54.2 | 50.2 |
| **Ours (with BiSAtt)** | 68.4 | 63.5 | 56.3 | 49.3 | 55.8 | 60.5 |
| **Ours (with CoSAtt)** | **68.7** | **63.8** | **57.0** | **51.5** | **56.0** | **61.9** |

Table 5: Performance of neural modules in our language-guided NMN implementation vs. state-of-the-art NMN models (on S-Ref test split).

find that neural modules `filter_ordinality`, `filter_visibility` significantly improve their performance with language guidance.

Additionally, we also compare the performance of models on C3-Ref+ contrast sets where the attributes are explicitly perturbed. Figure 1, Figure 2, Figure 3, Figure 4 show the performance of models IEP-Ref, FiLM, Vector NMN and NS-VQA respectively. Evaluation results using our approach are shown in Figure 5 and Figure 6. As we can see, majority of the models are robust to perturbations in color and shape indicating that these are relatively easier concepts to localize in the image. All the baselines including IEP-Ref show a significant drop of up to 20% on size, material, ordinality and visibility based perturbations. However, our language-guided model show relatively lower drop on these attributes, suggesting that our model is more robust to adversarial perturbations.

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

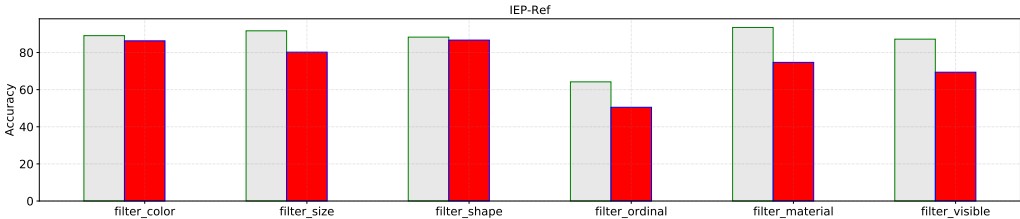

Figure 1: Performance of baseline IEP-Ref model on original test split (White bars) and C3-Ref+ contrast samples (Red bars)

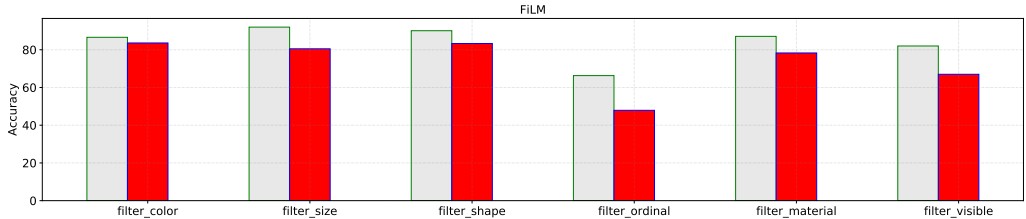

Figure 2: Performance of baseline FiLM model on original test split (White bars) and C3-Ref+ contrast samples (Red bars)

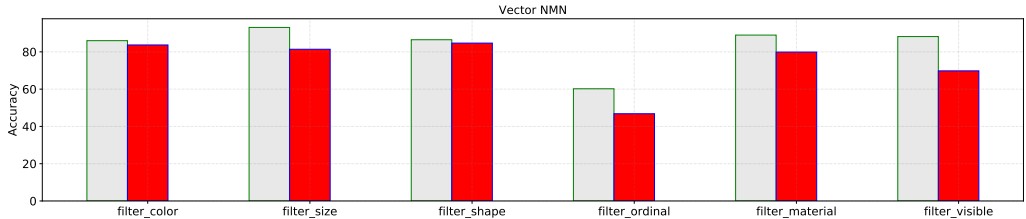

Figure 3: Performance of baseline Vector NMN model on original test split (White bars) and C3-Ref+ contrast samples (Red bars)

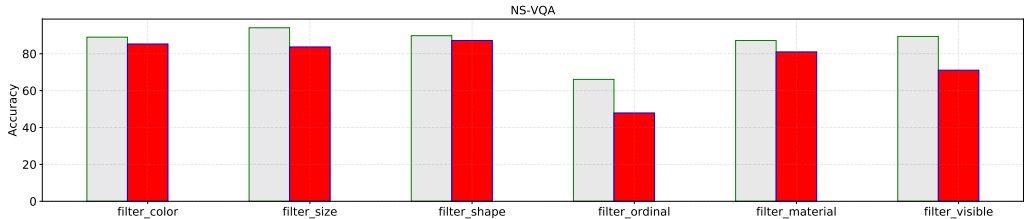

Figure 4: Performance of baseline NS-VQA model on original test split (White bars) and C3-Ref+ contrast samples (Red bars)

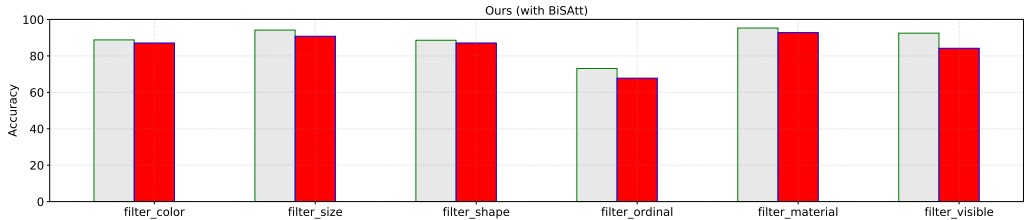

Figure 5: Performance of our approach (with BiSAtt encoder) on original test split (White bars) and C3-Ref+ contrast samples (Red bars)

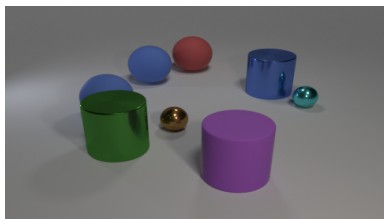
**C3-Ref+:** Any other large cylinder(s) that have the same material as the second one of the sphere(s) from left.

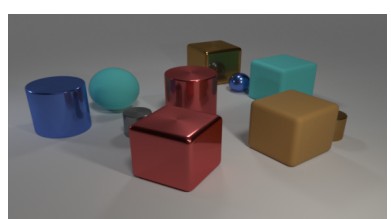
**C3-Ref+:** The sphere(s) that are both in front of the first one of the small cylinder(s) from left and have the same material as the second one of the big things from left.

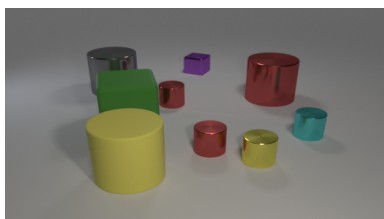
**C3-Ref+:** The rubber things that are either the sixth one of the thing(s) from right or the second one of the objects(s) from right that are behind the first one of the tiny cyan things(s) from right.

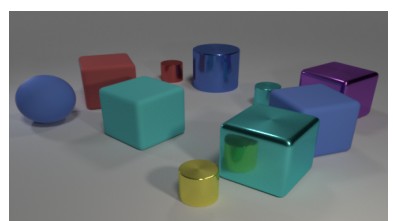
**C3-Ref+:** The metallic objects that are in front of the second one of the objects(s) from right that are of the same size as the third one of the metal thing from front.

Table 6: Random examples of novel compositions in C3-Ref+. The colors highlight the parts of the expressions obtained from different train samples in S-Ref.

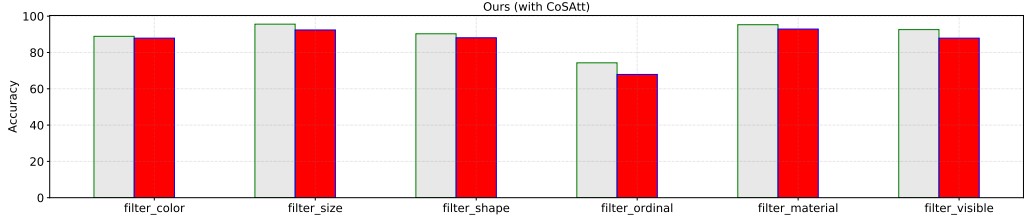

Figure 6: Performance of our approach (with CoSAtt encoder) on original test split (White bars) and C3-Ref+ contrast samples (Red bars)

[5] R. Liu, C. Liu, Y. Bai, and A. L. Yuille. Clevr-ref+: Diagnosing visual reasoning with referring expressions. In *IEEE Conference on Computer Vision and Pattern Recognition, CVPR 2019*, pages 4185–4194, 2019.

[6] J. Lu, D. Batra, D. Parikh, and S. Lee. ViLBERT: Pretraining task-agnostic visiolinguistic representations for vision-and-language tasks. In *NeurIPS*, 2019.

[7] E. Perez, F. Strub, H. de Vries, V. Dumoulin, and A. C. Courville. Film: Visual reasoning with a general conditioning layer. In *AAAI*, 2018.

[8] K. Yi, J. Wu, C. Gan, A. Torralba, P. Kohli, and J. Tenenbaum. Neural-symbolic vqa: Disentangling reasoning from vision and language understanding. In *Advances in neural information processing systems*, pages 1031–1042, 2018.

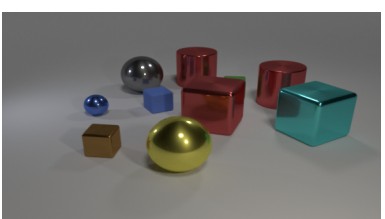
**Original:** Any other yellow metal thing(s) of the same size as the first one of the cyan metal thing(s) from right
**C3-Ref+:** Any other tiny thing(s) of the same size as the first one of the brown metal thing(s) from left

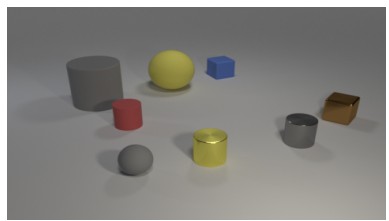
**Original:** The first one of the large rubber cylinder(s) from right that are in front of the thing that is behind the third one of the small matter object(s) from front
**C3-Ref+:** The first one of the small rubber cylinder(s) from right that are in front of the thing that is behind the third one of the small matter object(s) from front

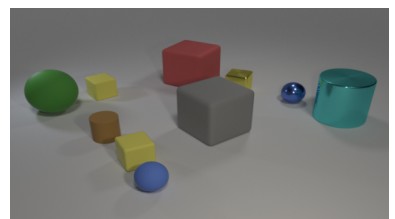
**Original:** The shiny thing(s) that are left of the first one of the small sphere object(s) from right and behind the fourth one of the rubber object(s) from front.
**C3-Ref+:** The shiny thing(s) that are right of the first one of the small cubical object(s) from right and behind the fourth one of the rubber object(s) from front.

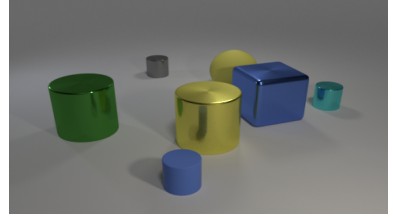
**Original:** Find matte thing that is on the left side of the cyan object that is behind the second one of the metallic object(s) from front; The last one of the object(s) from front that are behind it.
**C3-Ref+:** Find matte thing that is on the left side of the cyan object that is behind the second one of the metallic object(s) from front; The second one of the object(s) from right that are in front of it.

Table 7: Random examples of contrast sets in C3-Ref+ and their original annotations in S-Ref.