# OpenReview forum: "Robust Visual Reasoning via Language Guided Neural Module Networks"
_NeurIPS.cc/2021/Conference — NeurIPS 2021 Poster_

### Official Review · Reviewer_e2WV · 2021-07-05

**Rating:** 5
**Confidence:** 4

**Summary:**

This paper proposes a novel extension to the neural module network (NMN) to tackle vision-language tasks. The proposed method applies a language-guided convolution operator to the neural modules, which improves information exchange and fusion between the visual and language modes. The proposed framework is applied to various benchmarks in the visual question answering (VQA) and referring expression recognition (REF) domains. Experiment results demonstrate a superior performance of the proposed method.

**Limitations And Societal Impact:**

Limitations are briefly mentioned but not systematically discussed.
Social impact of this work is briefly discussed.

**Main Review:**

The capability to process complex information from multimodal perception is a hallmark of the power of modern deep learning systems. This paper focuses on the vision-language tasks, a direction currently subject to huge interest in both research and application. The proposed method introduces a way to further enhance the guidance to the reasoning modules by language inputs, and is claimed to be the first model to adapt convolutional filters to languages. Large amounts of carefully conducted experiments show improvements in performance over a wide range of baseline models on both the VQA and RFE tasks. The paper is well written and easy to follow, despite missing a few technical details which will help readers from outside the community to understand better.

My challenges and question to this paper are listed below. No further experiments are requested in the rebuttal.

1. The idea behind the design of both NMN models and the CLEVR dataset is to guide machines to learn "universal" neural modules that perform atomized logic operations through compositional reasoning. This means that the module parameters should not only be independent to the visual input, but also the language input. For example, in both "What size is the red sphere?" and "How many large red objects are there?" the "filter_color[red]" operator should map to the same logic operation regardless of the context input. This paper to some extend violates this design philosophy, therefore it becomes important to address how much the model is overfitted.

2. The language dependency of NMN mainly falls on the inference of programs. Is the language-guided execution modules able to fine-tune an imperfect program generator (for example, in VQA, first use a few thousand programs to train a weak program generator then jointly fine-tune with RL)? This might also suggest how overfitted the model is.

3. Any specific reason for choosing the Gaussian kernel? How do the results depend on the type of kernel and their sizes?

4. Since all guidance kernels are shared by all modules, would "modularizing" the guidance kernels further contribute to the performance?

Overall I think this paper has very strong potential and the experiments are solidly conducted. However, due to concerns about the model design, possible overfitting, limited novelty, I hereby give a "borderline reject" score.

**Time Spent Reviewing:**

2.5

---

> ### Author Response · Authors · 2021-08-10
> **Response to Reviewer e2WV**
>
> Thanks for your time and inputs! Please see the general response: https://openreview.net/forum?id=T1f0YKPP_K&noteId=6EuSAxlWI-c
>
> **Q: Model overfitting?**
>
> A: Through parametrization, our model uses significantly less parameters compared to other NMN models (Lines 161 - 164). Therefore, we believe our model is relatively less prone to overfitting. Our empirical results on multiple benchmarks and C3-Ref+ clearly demonstrate that our model is more robust and shows improved generalization capabilities in grounding unseen instances.
>
>
> **Q: Is the language-guided execution modules able to fine-tune an imperfect program generator?**
>
> A: This is an excellent question. We did not see any significant improvements in the performance of the program generator through jointly fine-tuning with Reinforcement Learning.
>
>
> **Q: Why Gaussian kernel?**
>
> A: The original PAC work only explored the Gaussian kernel as it has a fixed parametric form. This allowed them to efficiently implement PAC in 2D space alleviating the need for using hash tables and special lattice structures in high dimension. We closely followed their implementation as we did not want to deviate from their original implementation. We will add more analysis in the final version by comparing results obtained using different types of kernel and sizes.
>
>
> **Q: Modularizing the guidance kernels?**
>
> A: We performed this experiment and we see a significant drop in model performance on all the benchmarks with modularizing the guidance kernels. Specifically, compared to our best model CoSAtt, we see 6.3% absolute percentage drop on CLOSURE, 4.1% absolute percentage drop on S-Ref Test, 1.4% absolute percentage drop on F-Ref Test, 2.3% absolute percentage drop on Cops-Ref and 13.6% absolute percentage drop on C3-Ref+.
>
>
> **Q: Limited Novelty?**
>
> A: Our work is amongst the first to study the limitations of hard-coded inputs and the impact of contextualizing NMN models in solving the REF task. Such an analysis is important and valuable to drive further research on modularized neural architectures like NMN.

---

### Official Review · Reviewer_tcpV · 2021-07-13

**Rating:** 7
**Confidence:** 3

**Summary:**

The authors introduce three new tools to improve compositionality in visual reasoning tasks:
1. They apply pixel-adaptive conv nets [Su et. al., 2019] to the modules of Neural Module Networks. The guidance features, which weight the convolution computation, are computed via attention over the full image and text query. They present two different attention architectures to compute this feature: one separates the attention computation over vision and language and the other co-attends.
2. A new CLEVR test set that is constructed from the S-ref split of CLEVR. The new test set consists of (a) object, spatial relationships not seen during training and (b) contrast sets [Gardner et. al., 2020].
3. They parameterize modules by argument (e.g., `blue`) to reduce the number of modules to be learned.
Their model achieves competitive performance on CLEVR and improves over all baselines on the S-Ref and F-Ref splits. On their new split (C3-Ref+) they achieve the lowest drop in accuracy and maintain performance as the number of comparison relationships stays high.

Pixel-Adaptive Convolutional Neural Networks. Su et. al., 2019.
Evaluating Models’ Local Decision Boundaries via Contrast Sets. Gardner et. al., 2020.

**Ethical Concerns:**

After reviewing the ethics guidelines, I believe there are no issues.

**Limitations And Societal Impact:**

In my view, the authors have adequately addressed the impact of their work.

**Main Review:**

Originality: The paper brings together a variety of modeling techniques for the first time: an adaptive convolution together with attention computations over image and text queries all inside a neural module network.

Quality: The evaluation was thorough, but some of the design decisions left me puzzled. The authors show results across a variety of splits, including a new split designed to probe robustness and appropriately ablate every aspect of their system. Their new system leaves me puzzled because it seems to break the benefits of neural modules by incorporating contextual input into the modules. If the benefit of the LG-Conv layer is to add textual relevance and whole image context, why use modules at all? For example, in their qualitative results, they show that their `filter_color(red)` module  better localizes the red cylindrical object in the scene compared to IEP-Ref, which just localizes every red object in the scene. Conceptually, the latter seems _more_ correct. The contribution seems to come more from the attention architectures rather than their interaction with the modules. To understand the contribution of attention further, it would be helpful for the authors to describe how ViLBERT and VisualBERT were applied to CLEVR and contrast it with their approach.

Clarity: The paper was clear when discussing background and experiment design. I found the method and motivation a little confusing. I also had to rely on the original pixel-adaptive conv paper to fully understand their changes (details under comments), but that did not affect the overall quality.

Significance: The paper puts forth a fine-grained analysis of a large number of methods interacting together with the goal of modularity. These insights could be useful in developing more compositional modules across a variety of visual tasks.

Questions:
- Equation (4): why is the optimization described as policy gradient? Is standard cross-entropy not appropriate?
- Are Visual BERT and ViLBERT given whole image information or are they restricted to object features extracted from the image?

Comments to authors (not considered in review score):
- The indexing in equation (2) is confusing. It would help to specify that these indices are 2-dimensional and to state explicitly that [] is an index operation.
- L201, Table 1: ViLBERT and Visual BERT resolve to the same reference

**Time Spent Reviewing:**

4

---

> ### Author Response · Authors · 2021-08-10
> **Response to Reviewer tcpV**
>
> Thanks for your time and encouraging review! Please see the general response: https://openreview.net/forum?id=T1f0YKPP_K&noteId=6EuSAxlWI-c
>
> **Q: How ViLBERT and VisualBERT were applied to CLEVR?**
>
> A: We closely followed the experiment setup used in  state-of-the-art works for training and evaluating multi-modal transformers (Lu et al., 2019, Li et al., 2020). We represent the input image as a set of image regions by extracting bounding boxes. The image region features are generated from a pre-trained Faster R-CNN object detection network. For all the synthetic datasets, we additionally fine-tuned the Faster R-CNN on the CLEVR train data.
>
> Lu et al., Vilbert: Pretraining task-agnostic visiolinguistic representations for vision-and-language tasks. In NIPS 2019
>
> Li et al., VisualBERT: A Simple and Performant Baseline for Vision and Language. In ACL 2020
>
> **Q: More background on the original pixel-adaptive conv paper?**
>
> A: Thanks for pointing this out. We will provide more detailed background on PAC operation in our final version.
>
>
> **Q: Optimization described as policy gradient in Equation 4?**
>
> A: The notion of rollout of the policy in the program generator (PG) and the execution engine (EE), and the lack of ground-truth programs for datasets containing real images, makes policy gradient to be a natural choice. Moreover, this helps in jointly optimizing both PG and EE without the need for a large number of ground-truth programs.
>
>
> **Q: Indexing in Equation 2 and Typo in Reference?**
>
> A: Thanks for pointing this out. We will address this in our final version.

---

### Official Review · Reviewer_vcX9 · 2021-07-16

**Rating:** 5
**Confidence:** 3

**Summary:**

The paper presents an incremental NMNs with a language-guided adaptive convolution layer into vanilla NMN. The goal o LG-Conv is to have a module that can adaptively co-attend over potential objects of interest from the visual input and textual input by altering the convolution.

To achieve this, they present biSAtt and CoSAtt networks to learn the so-called guidance kernel G from textual and visual inputs.

Experiments are done on multiple widely accepted visual reasoning benchmarking datasets, as well as a new benchmark that is designed to explicitly test the model's ability to generalize to adversarial perturbations.


**Ethical Concerns:**

None.

**Limitations And Societal Impact:**

A brief broader impact narrative is provided. I do not see any explicit potential negative societal impact of this work.

**Main Review:**

The power of NMNs were argued as leveraging specialized modules and could be assembled to perform complex and compositional reasoning. The decomposed module decouples the visual pathway and language pathway, forming an elegant structure. The argument of this work to me violates this design elegancy by agglomerate language signal again back into the convolution operations. The fact of achieving high performance on CLEVR-ref and others, as shown by the authors, could also be interpreted  CLEVR like dataset is way too artificial that could not reflect the power of NMNs following its original designing principle.

This is being shown in Figure 4. The filter_color(red) is designed to be a modular operator only to locate red thing but not attending to other so-called objects with contextual relevance. By arguing the filter_color(red) should also consider contextual relevance goes against the NMNs principle.

If an incremental implementation does not follow a vanilla architecture's original designing principle but achieves better performance, how shall we as a community interpret it?

My other concern is with the CLEVR-Ref+ construction. For example, in figure 5, both E1 and E4 refers to an object that is to the left of yellow thing (and there is only one yellow thing). However, the GT for E1 is r1, and E4 is r4, one is to the left, and the other is to the right. My interpretation could be wrong, but from this single example, there seems to be inconsistent GT already. The authors mentioned three human annotators validate the creation of the Ref+ dataset. However, if the dataset fidelity is not well maintained, all the experimental results won't be convincing for making solid arguments.

Furthermore, one argument of this work is "NMNs fail to understand new concepts such
as “yellow sphere to the left" even when it is a combination of known concepts
from train data: “blue sphere", “yellow cube", and “metallic cube to the left"."  I am wondering if the authors have verified in the new Ref+ dataset what is the percentage of such a combination of known concepts? How would you make sure when the human labeler is editing the phrases, they will create new ones that were not in other examples?

Other places need clarification:
In some places, you use NMNs. Some use NMN. Would you please make sure to unify them?
Figure 5, E3 is constructed from E1 and E2, but having a new phrase, "big blue rubber thing" instead of "big blue metallic thing". Are you also perturbing these phrases manually? Also, could you explain why the GT of E2 is <box:r2>. It doesn't seem to be correct to me.


**Time Spent Reviewing:**

3 hours

---

> ### Author Response · Authors · 2021-08-10
> **Response to Reviewer vcX9**
>
> Thanks for your time and inputs! Please see the general response: https://openreview.net/forum?id=T1f0YKPP_K&noteId=6EuSAxlWI-c
>
> **Q: E3 and E4 are incorrect in Figure 5?**
>
> A: Our C3-Ref+ annotations, as verified by three human annotators, are very clean and validated. The annotations E3 and E4 shown in Figure 5 are correct. E1 and E4 do not refer to the same object. Please note that there are two yellow things in the image. The second yellow object, i.e. the *‘small metallic yellow cube’* is behind all the objects in the image and is partially visible.
>
>
> **Q: Percentage of combination of known concepts in the newly constructed C3-Ref+ dataset?**
>
> A: In constructing our C3-Ref+ dataset, we first automatically extract all the object phrases (such as *'yellow sphere to the left'*, *'tiny metallic thing on the right'*, etc) using simple parts of speech tags based templates. We then automatically filter out all the combinations that are not in the train data. These combinations are then used by our annotators to construct new examples manually. Therefore, all the examples in our C3-Ref+ dataset contain at least one combination not seen in the train data.
>
>
> **Q: NMNs vs NMN?**
>
> A: Thanks for pointing this out. We will use a consistent notation in our final version.
>
>
> **Q: CLEVR like dataset is way too artificial/synthetic that could not reflect the power of NMNs?**
>
> A: We also showed improvements on the Cops-Ref dataset, which is not a synthetic dataset.

---

> > ### Comment · Reviewer_vcX9 · 2021-09-11
> > **I found the second yellow object in Figure 5!**
> >
> > Thank you for pointing out the second yellow metallic object in figure 5... I didn't notice that... and also showing how hard the task is for human annotators.
> >
> > Yet, in this case, then r1 and r4 are both correct answers to both E1 and E4?
> >
> > To E4, r1 is a metallic object (verified by having r1 as gt for E1) and behind the metallic cylinder (both r1 and r4 are), and to the left of the yellow thing (the partially occluded one).
> >
> >     > <

---

> > > ### Author Response · Authors · 2021-09-13
> > > **Thanks for the question!**
> > >
> > > Thanks for the question. We have verified our annotations and find that two of the three annotators find r4 as the correct annotation for E4, probably because the reference objects used in E4 such as metallic cylinder and the yellow thing are more relatively closer to r4 compared to r1. However, as pointed out by the reviewer, it is possible to argue that r1 is also a correct annotation for E4.
> > >
> > > We would like to point out that the ambiguity here arises due to the ambiguity in the original dataset for E1 (and this is very rare as the original annotations are verified through multiple validation steps).  As we construct contrast sets on top of existing annotations, some ambiguity could have been propagated into our contrast sets. We will perform another round of validation to remove these ambiguous annotations in our final release set.
> > >
> > > We hope our responses have also clarified any of your remaining concerns. Please consider updating the score if your concerns are addressed or let us know of any remaining issues. Thanks.

---

### Official Review · Reviewer_Sgvd · 2021-07-17

**Rating:** 7
**Confidence:** 2

**Summary:**

This paper introduces a language-guided adaptive convolution layer into neural module networks (NMN) to address the limited generalizability of NMN to unseen concepts. The proposed method is able to achieve on-par performance to state-of-the-art models on CLEVR VQA task while significantly outperforms state-of-the-art models on CLEVR-Ref+ and CopsRef.  The model is also able to generalize better to novel compositions of objects and relations in CLOSURE.

In addition, a new challenging out-of-distribution test split for REF task is introduced, to evaluate on model generalizability to adversarial perturbations and unseen combinations of known concepts. The proposed method has shown a clear advantage on this new test split.

**Limitations And Societal Impact:**

Yes.

**Main Review:**


Overall, the paper is well written, with extensive experiments conducted on a comprehensive list of datasets and tasks. The performance improvements from the proposed model are significant, especially on several referring expression recognition tasks.

In addition, the proposed dataset extends existing datasets of testing model generalizability to another dimension: adversarial perturbation. It can be potentially widely adopted by future works along this direction.

**Time Spent Reviewing:**

2

---

> ### Author Response · Authors · 2021-08-10
> **Response to Reviewer Sgvd**
>
> Thanks for your time and encouraging review!

---

### Author Response · Authors · 2021-08-10
**General Response to Reviewers**

We thank all the reviewers for their thoughtful feedback and constructive comments. We are delighted to hear that the reviewers think we addressed an important (e2WV), well-written (Sgvd, e2WV), clear (tcpV, e2WV), interesting problem (e2WV) with solid (e2WV), significant empirical results (Sgvd), extensive, thorough (tcpV) experiments conducted on a widely accepted (vcX9), comprehensive list of datasets and tasks (Sgvd, tcpV, e2WV); and that our work brings together a variety of original (tcpV), novel (e2WV) modeling techniques for the first time and also appropriately ablate every aspect of our proposed model (tcpV). Reviewers mention that our work is easy to follow (e2WV), has very strong potential (e2WV) and the insights drawn in the paper could be useful in developing more compositional modules. In addition, Reviewers find our proposed C3-Ref+ dataset to be very useful and can be potentially widely adopted by future works along our research direction (Sgvd). We will incorporate all feedback in the final version.


**Q: Violation of NMN’s Design Principles?**

A: This is an excellent question. Our idea of adding parametrization and contextualization to NMN can be viewed as the addition of *conditional normalization* (CN) to NMN. CN is a proven, highly effective, and a popular line of research work in deep learning, for example Conditional Batch Norm (De Vries et al., 2017), Conditional Instance Norm (Dumoulin, et al., 2017), and Adaptive Instance Norm (Huang et al., 2017). Specifically, conditional normalization on language, visual inputs is shown to improve grounding complex visual scenes in multi-modal benchmarks such as VQA (De Vries et al., 2017).

Moreover, the neural modules in our architecture are still modularized and perform filtering based on their logical representation/meaning. For example, *filter_color('red')* only localizes the red colored objects in the image (for example, in figure 4) and does not localize other objects such as yellow colored objects or metallic objects. This does not deviate from the original design principle of NMN where the goal is to have a dedicated/specialized/interpretable neural module for performing a particular semantic/logical operation. With added contextualization, our *filter_color('red')* module learns to filter out less relevant red colored objects, but still localizes only red colored objects. We believe that this does not violate the modularity and interpretability design aspects of NMN but rather improve the grounding abilities in complex scenes (as illustrated with our results on the Cops-Ref dataset containing real images with complex objects).

Finally, our experiment results on both real (Cops-Ref) and synthetic (CLEVR-REF+, CLOSURE) REF datasets demonstrate the importance of parametrizing neural modules in NMN and the impact of contextualizing these modules in improving generalization and robustness to adversarial perturbations.

[1] De Vries et al., Modulating early visual processing by language. In NIPS 2017.

[2] Dumoulin, et al., A learned representation for artistic style. In ICLR 2017.

[3] Huang et al., Arbitrary style transfer in real time with adaptive instance normalization. In ICCV 2017.

To reiterate, our main contributions are:

1) Our work is amongst the first to study the limitations of hard-coded inputs and the impact of contextualizing NMN models in solving the REF task.

2) We conduct extensive experiments on both real and synthetic REF datasets to show the evidence that contextualization is the key in improving grounding abilities of NMN.

3) We achieve new state-of-the-art results on multiple tasks and benchmarks.

4) We introduce a new benchmark to explicitly test the model’s ability to generalize to adversarial perturbations and novel compositions of concepts not seen during training. We show that our model, through parametrization, uses less parameters overall, making it less prone to overfitting and more robust and generalizable compared to previous approaches.

---

### Author Response · Authors · 2021-08-31
**Reg: Thank you Reviewers!**

We thank reviewers again for their valuable feedback. We hope our responses addressed several of the reviewers concerns. Please let us know if there are any more concerns or questions. Thanks.

---

### Decision · Program_Chairs · 2021-09-27

**Decision:**

Accept (Poster)

**Comment:**

The paper proposes a new model for compositional reasoning with extensive evaluation on synthetic and  real image datasets. Additionally, the paper contributes a new challenging ref test set based on CLEVR which is manually verified.

Given the interesting results and technical contributions the paper provides, I recommend to accept this paper under the expectation that the authors will revise the paper as promised in the author response, including:
1) Fix notational issues / references, consistent notation
2) Additional analysis using different types of kernel and sizes
3) Adding results on Modularizing the guidance kernels

Several reviewers raised the concern how "modular" this design is and how good the grounding is. I encourage the authors to include more of the discussion they provide in the author response to the final version of the paper. I think it would be very convincing to see a quantitative analysis on this, on the precision (and maybe recall for completeness as well) of the attention, i.e. a quantify the analysis of Figure 4, ideally comparing to baselines and prior work. I believe this should be reasonable easy given the annotations provided with CLEVR, and significantly increase the insight of this work and thus increase its impact.